# Early Palliative Care in Patients with Glioblastoma: Co-Design of an Integrated Care Pathway

**DOI:** 10.3390/healthcare13182378

**Published:** 2025-09-22

**Authors:** Esmée K. J. van der Poort, Monique C. M. Baas-Thijssen, Marleen Oomes, Maaike J. Vos, Robin M. Pieterman, Martin J. B. Taphoorn, Inge de Vries, Carla Juffermans, Eline F. de Vries, Yvette M. van der Linden, Johan A. F. Koekkoek

**Affiliations:** 1Department of Neurology, Leiden University Medical Center, 2333 ZA Leiden, The Netherlandsr.m.pieterman@lumc.nl (R.M.P.);; 2Center of Expertise in Palliative Care, Leiden University Medical Center, 2333 ZA Leiden, The Netherlands; 3Department of Neurology, Haaglanden Medical Center, 2512 VA The Hague, The Netherlands; 4Zorg & Zekerheid, Regional Health Insurance Company, 2332 KG Leiden, The Netherlands; 5Department of Public Health and Primary Care, Leiden University Medical Center, 2333 ZA Leiden, The Netherlands; 6Department of Radiotherapy, Leiden University Medical Center, 2333 ZA Leiden, The Netherlands

**Keywords:** palliative care, integrated care, glioblastoma, advance care planning

## Abstract

**Background:** Glioblastoma is an incurable form of brain cancer with a median overall survival of 1.5 years. Despite its progressive nature and high symptom burden, palliative care is not consistently integrated in routine glioblastoma care. Early integration of palliative care better addresses the needs of patients and caregivers, improves quality of life, and reduces inappropriate care in the end-of-life phase. This study aims to design an integrated care pathway to support the early integration of palliative care for patients with glioblastoma. **Methods:** We used a design thinking approach, engaging stakeholders from neuro-oncology, specialist palliative care, primary care, district nursing, healthcare administration, health insurance, health economics, and patient advocacy. The process consisted of thirteen informal interviews (with healthcare professionals, patients, and caregivers), six expert meetings, and two workshops. **Results:** First, we mapped existing routine glioblastoma care and identified perceived barriers to early palliative care integration, including variations in advance care planning (ACP) timing, clinicians’ hesitation, unclear referral criteria to specialist palliative care, suboptimal care coordination, and limited experience with glioblastoma in the primary care setting. Second, iterative prototyping led to the development of a care pathway with key components: initiation of ACP by the lead clinician within six weeks of diagnosis, integrated multidisciplinary team meetings for complex cases, ongoing coordination, clear referral triggers for specialist palliative care, and structured caregiver care. **Conclusions:** The co-designed pathway provides a feasible model for integrating early palliative care into routine care for patients with glioblastoma. Future steps include implementation and evaluation of the care pathway and development of a payment model.

## 1. Introduction

Glioma is an incurable form of primary brain cancer which affects approximately 6 in 100,000 persons annually worldwide [1,2]. Around 80% of patients are diagnosed with glioblastoma, the most severe subtype of glioma. Despite multimodal treatment with surgery, radiotherapy, and chemotherapy, patients have a limited median overall survival of approximately 1.5 years [3].

Patients with glioblastoma do not only have cancer but also suffer from progressive neurological impairments, related to the tumor location, which impact, for example, their memory, behavior, and/or ability to communicate [4]. These progressive neurological symptoms may reduce patients’ medical capacity to engage in treatment decisions, complicating the care of patients with glioma. The impact of the disease extends beyond patients to their family and friends, who frequently take on increased caregiving responsibilities. As a result, caregivers often report heightened levels of stress and emotional burden [5].

Given the progressive and incurable nature of glioblastoma, the integration of early palliative care into routine care is essential to support treatment decision-making and optimal symptom management [6]. Integrated cancer care should follow two tracks: a disease-modifying track focused on cancer treatment and a palliative care track focused on quality of life [7]. A key component of integrated palliative care is advance care planning (ACP). ACP is “a process that supports adults at any age or stage of health in understanding and sharing their personal values, life goals, and preferences regarding future medical care. The goal of ACP is to help ensure that people receive medical care that is consistent with their values, goals and preferences during serious and chronic illness” [8]. Previous studies indicate that ACP improves symptom management and health-related quality of life (HRQoL), while also reducing inappropriate care in the end-of-life phase [9,10].

Despite the potential benefits of early integration, palliative care is not consistently embedded in routine clinical practice [11]. Care delivery for patients with glioblastoma generally remains fragmented, with limited coordination between healthcare providers, resulting in suboptimal ACP, inadequate symptom management and insufficient psychosocial support [12,13,14]. These gaps underscore the pressing need to systematically integrate palliative care into routine neuro-oncological care. This study aims to design an integrated care pathway to support the early integration of palliative care for patients with glioblastoma. The proposed pathway aims to foster better interprofessional collaboration, reduce unnecessary interventions, and improve overall HRQoL for both patients and their caregivers.

## 2. Methods

### 2.1. Context

The present study is part of a broader initiative exploring payment reform of palliative care in the Netherlands. Commissioned by the Dutch Health Care Authority (the national regulatory body responsible for oversight and regulation of the healthcare market) and the Netherlands Organization for Health Research and Development the initiative comprises three experimental projects aimed at developing alternative payment models for palliative care. The current experiment focuses specifically on patients with glioblastoma. It follows a two-phase design: first, the development of an integrated care pathway, and subsequently, the design of an alternative payment model. This study reports on the first phase: the co-design of the integrated care pathway.

The integrated palliative care pathway was developed in an urban area of South Holland province in the Netherlands. In the Netherlands, 1300 patients are diagnosed with glioma annually [15] and an estimated 150 to 200 cases occur in the South Holland area. Hospital-based care for patients with glioma is delivered through a network of specialized centers, which collaborate with regional hospitals via integrated oncology networks. GPs play a central role in coordinating care in the home setting in close collaboration with district nursing. A key feature of the Dutch healthcare system is that nearly all citizens are registered with a general practitioner (GP).

### 2.2. Participants and Governance

The design process started by identifying all key stakeholders in day-to-day glioblastoma care in the region under study. Key stakeholders from the fields of neuro-oncology, specialty palliative care, primary care, district nursing, healthcare administration, health insurance, health economics, and patient advocacy were engaged. These stakeholders were invited to join the project group and were actively part in the design of the care pathway. Furthermore, each organization in the project group selected one representative to join the steering committee of the project.

The design process consisted of several types of recurring meetings. The project leaders met biweekly to monitor progress, plan upcoming meetings and workshops, and coordinate engagement of stakeholders. The steering committee met every four weeks and oversaw final decision-making and facilitated collaboration across participating healthcare providers. In addition, the project group participated in two plenary sessions. Representatives from hospital care, primary care, and district nursing additionally participated in two co-design workshops.

Between September 2024 and June 2025, a total of 27 stakeholders participated in the design process. The steering committee consisted of 11 members who met during six steering committee meetings. The steering committee and four additional project group members additionally participated in two plenary sessions and two co-design workshops. At least one patient advocate was present for all steering committee meetings and plenary sessions. Additionally, five healthcare providers, two patients and two caregivers were consulted during informal interviews in the first phase of the project. An overview of all involved stakeholders and their roles is provided in Table 1.

### 2.3. Design Thinking

A design thinking approach was used to guide the development of the integrated care pathway. Design thinking is a user-centered methodology that helps address complex problems by designing solutions in close collaboration with relevant stakeholders [16,17]. Design thinking is an iterative process structured around five steps: empathize, define, ideate, prototype, and test. The empathize step focuses on gaining a deep understanding of users’ experiences and challenges. During the define step, these insights are synthesized into a clear, user-centered problem statement. The ideate step involves generating a broad range of potential solutions, followed by the prototype step, where prototypes are refined based on early feedback. By following these steps, design thinking supports continuous co-creation and refinement with stakeholders. In this study, we report on four of the five steps, structured into two phases: (1) the current care pathway (empathize and define) and (2) the ideal integrated care pathway (ideate and prototype). Figure 1 shows a schematic timeline of the project with the two design thinking phases and project meetings.

#### 2.3.1. Phase 1: Current Care Pathway (Empathize and Define)

The first phase focused on mapping the existing care pathway and identifying perceived barriers to the early integration of palliative care in routine glioblastoma care. The initial plenary session, held in October 2024, aimed to foster collaboration among stakeholders. During this session, each stakeholder gave a brief presentation outlining their role in glioblastoma care and their potential contribution to the project. Between November 2024 and January 2025, six recurring steering committee meetings were held to reflect on the structure of the current care pathway and discuss perceived barriers to early palliative care integration.

When specific perspectives were found to be underrepresented, the project leaders, under guidance of the steering committee, invited additional stakeholders for informal interviews to reflect on perceived barriers to the early integration of palliative care. Patients and caregivers were initially approached by the participating patient advocate through the patient association for glioblastoma in the Netherlands. When patients and/or caregivers agreed to participate, they were contacted by the main researcher (E.K.J.v.d.P) for an informal interview. At the start of the interview, they were informed about the project and that the interview has no effect on the care they receive, and they were also asked for verbal informed consent. The interviews were not recorded but the researcher took notes, which were send to the patients and caregivers after the interview for their review and approval. In total, two patients and two caregivers participated in an informal interview. Healthcare professionals of relevant backgrounds not included in the project group were approached by steering committee members and when they agreed to participate were contacted by the researcher. Participants were informed that participation in informal interviews implied consent for the anonymous use of interview notes for research purposes. Besides the two patients and two caregivers, five additional healthcare professionals and four steering committee members participated in informal interviews (total of 13 interviews).

Qualitative data were collected in the form of minutes, field notes, project documentation and informal interview summaries. Qualitative data were analyzed in Word using thematic analysis to identify perceived barriers. Analysis took place in an iterative process, where identified barriers would be discussed in steering committee meetings and refined.

#### 2.3.2. Phase 2: Ideal Integrated Care Pathway (Ideate and Prototype)

The second phase of the project focused on the development of an ideal integrated care pathway through iterative ideation and prototyping. Building on the perceived barriers identified in Phase 1, stakeholders collaboratively defined project success criteria, distinguishing between essential (need-to-have) and desirable (nice-to-have) features of the care pathway. The ideation process was informed by the Lynn and Adamson framework for palliative care [18] and supplemented by relevant best practices in the field [19,20]. Based on these inputs, the project leaders drafted an initial version of the integrated care pathway, which was refined through two co-design workshops. The first workshop, held in January 2025, included two healthcare professionals from neuro-oncology and two specialist palliative care, while the second, in March 2025, expanded participation to two general practitioners and one district nursing professional.

The care pathway was further refined through discussions during recurring steering committee meetings, where stakeholders provided feedback on content, feasibility, and implementation. In parallel, stakeholders selected indicators to monitor and evaluate the outcomes, quality, and costs of integrated palliative care. A plenary session in February 2025 was dedicated to addressing structural barriers in healthcare financing and exploring options for a supporting payment model.

### 2.4. Ethical Considerations

This study was conducted as a care evaluation, aimed at improving the quality and organization of routine care without subjecting participants to any treatment or intervention. As such, this study did not fall under the scope of the Dutch Medical Research Involving Human Subjects Act (WMO).

## 3. Results

### 3.1. Phase 1: Current Care Pathway (Empathize and Define)

The steering committee first identified key challenges in the delivery of palliative care for patients with glioblastoma. These challenges were consequently discussed with additional healthcare providers, patients, and caregivers during informal interviews. This resulted in four core problem statements: (1) ACP is often initiated late in the disease trajectory (frequently in the final three months of life) or not at all, (2) palliative care is fragmented, with poor coordination between healthcare providers, (3) patients may receive potentially inappropriate care in the end-of-life phase that does not align with their values and preferences, and (4) bereavement care for family members is not consistently provided. Stakeholders were then asked what barriers contribute to these challenges. These perceived barriers (spanning clinical practice, digital infrastructure, and financing) are summarized in Table 2 alongside the corresponding problem statements.

Regarding the timing of ACP, stakeholders emphasized the complexity of starting ACP conversations. Each patient has individual preferences and ACP discussions can be emotionally challenging to patients and caregivers. Simultaneously, they noticed that healthcare professionals may feel uncertain about how and when to initiate such conversations. At the onset of the study, ACP consultations in the hospital were not reimbursed, potentially limiting the time clinicians would dedicate to this critical aspect of palliative care. Additionally, the absence of a shared digital infrastructure made it difficult for providers across different healthcare settings to collaborate and exchange care plans. As a result, neuro-oncologists, general practitioners, and district nurses were often unaware whether another healthcare professional had initiated ACP and which topics were discussed with the patient.

During stakeholder discussions, it became clear that the described challenges and barriers are often interrelated. For example, a shared data infrastructure may improve the coordination between providers and, consequently, soften the transition between the secondary and primary care setting. In addition, timely initiation of ACP and better coordination between providers could contribute to a reduction of potentially inappropriate care in the final phase of life.

A key point of the stakeholder discussions was the relative rarity of glioblastoma in the GP practice. Depending on the geographic location, GPs may encounter only one patient with glioblastoma in their practice over the course of their entire career. As a result, primary care providers often lack familiarity with the disease and may feel uncertain about the management of progressive neurological symptoms. Both GPs and district nurses emphasized the need for targeted education and improved communication in the final phase of life with the hospital-based neuro-oncology team.

### 3.2. Phase 2: Ideal Integrated Care Pathway (Ideate and Prototype)

The definitive integrated care pathway for palliative care in patients with glioblastoma is shown in Figure 2. The integrated care pathway is structured into layers, each representing a key aspect of palliative care during the disease trajectory for patients with glioblastoma. The first layer shows the palliative care model by Lynn and Adamson [18] and outlines five distinct phases. Diagnosis includes diagnostic imaging and obtaining a histopathological diagnosis following biopsy or tumor resection. Disease-modifying care refers to treatment aimed at prolonging life without curative intent, such as chemo- and/or radiotherapy. In parallel, symptom management focuses on relieving and managing symptoms to preserve HRQoL. These two types of care often occur simultaneously and gradually shift in emphasis as the disease progresses. The dying phase marks a transition in focus from quality of life to quality of dying, typically encompassing the final days of life. Following the patient’s death, aftercare is provided to support bereaved family members.

The well-being layer captures the patient’s burden of suffering across physical, psychological, social, and spiritual domains throughout the palliative phase, which is continually monitored using patient-reported outcome measures (PROMs). PROMs include the European Organization for Research and Treatment of Cancer (EORTC) Quality of Life Questionnaire Core 30 (QLQC30) [21], the EORTC Brain Cancer Module (BN20) [22], the EORTC Instrumental Activities of Daily Living in brain tumors patients (IADL-BN32) [23], the Hospital Anxiety and Depression Scale (HADS) [24], and an epilepsy questionnaire to measure seizure control. The figure shows a conceptual overview of progressively worsening well-being during the disease trajectory.

The initiation of palliative care layer marks the start of the palliative phase at diagnosis and outlines specific moments in the disease trajectory for scaling up of palliative care. At diagnosis, the lead clinician, together with the nurse practitioner, will conduct conversations on ACP and begins implementing palliative care measures. As patients’ preferences may evolve over time, ongoing communication and shared decision-making between patient and the clinical team are essential to guide subsequent care decisions. Key moments for scaling up palliative care were defined at the following time points in the disease trajectory: diagnosis or treatment initiation, disease progression or clinical deterioration, when the patient no longer desires anti-tumor treatment, transfer to home care, and, lastly, the onset of the dying phase. Other incidental occasions for scaling up palliative care are the request of the patient or caregiver, high caregiver burden, and an increase in symptom burden according to the clinician or experienced by the patient and/or relatives.

Hospital care includes the various roles of specialized providers involved in neuro-oncological care, such as neurologists, neuro-oncologists, nurse practitioners, neurosurgeons, medical oncologists, radiation oncologists, and palliative care specialists. The hospital palliative care consultation team is also often involved. In contrast, the home care layer reflects the roles of general practitioners, district nurses, and palliative care nurses, who support patients in the home environment.

The collaboration layer highlights the importance of coordinated activities that link hospital-based and home-based care providers, ensuring continuity and alignment across settings. The care coordination layer defines the lead clinician responsible at each stage of the care process. Typically, the neuro-oncologist leads during active treatment and palliative care will be provided in the hospital setting, while this role is transferred to the GP once anti-tumor treatment is no longer pursued and palliative care will be delivered in the home setting.

The supportive services layer refers to practical and psychosocial support provided in the home, including domestic assistance, home modifications, and support for family members and informal caregivers.

### 3.3. Monitoring and Evaluation

As part of the future testing and implementation of the integrated care pathway, ongoing monitoring of care quality, outcomes, and costs were viewed to be essential to support continuous improvement in care delivery. Stakeholders collaboratively defined four quantifiable short-term goals aligned with the four core problem areas, intended to track the performance of the pathway over time: (1) in at least 80% of patients with high-grade glioma, ACP is carried out within 3 months of diagnosis, (2) in at least 80% of patients, a central integrated care coordinator is appointed, (3) in at least 80% of patients, care in the end-of-life phase is appropriate as defined by De Schreye et al. [25]), and (4) in at least 80% of bereaved individuals, aftercare is provided within 3 months. Note that these goals may be adjusted depending on the current status of, for example, appropriate care in the end-of-life phase.

These short-term targets were defined as intermediate steps toward achieving the overarching, long-term goal of improving the quality of care. Additional long-term objectives include increased satisfaction among patients and their caregivers, enhanced collaboration and job satisfaction among all involved healthcare providers, and a reduction in unnecessary healthcare expenditures. To support evaluation, a core indicator set was developed in collaboration with stakeholders (Table 3). This set will be used to monitor the performance and impact of the care pathway over time and to inform future refinement and scaling.

## 4. Discussion

In this study, we aimed to design an integrated care pathway for patients with glioblastoma using a design thinking approach. Collaborating with stakeholders from patient advocacy, neuro-oncology, palliative care, general practice, district nursing, healthcare administration, health economics and health insurance, we co-created an integrated care pathway that addresses key barriers to the early integration of palliative care. First, we mapped existing routine glioblastoma care and identified perceived barriers to early palliative care integration, including variations in advance care planning (ACP) timing, clinicians’ hesitation, unclear referral criteria to specialist palliative care, suboptimal care coordination, and limited experience with glioblastoma in the primary care setting. Second, iterative prototyping led to the development of a care pathway with key components: initiation of ACP by the lead clinician within six weeks of diagnosis, integrated multidisciplinary team meetings for complex cases, ongoing coordination, clear referral triggers for specialist palliative care, and structured caregiver care. The present model aims to promote timely initiation of palliative care by using the two track approach, including ACP, and to foster interprofessional collaboration among key healthcare professionals involved in glioblastoma care, both in the hospital setting and at home.

### 4.1. Integrating Palliative Care into Routine Oncological Care Early

Both international and national guidelines, including those from the American Society of Clinical Oncology ASCO and the Dutch Oncology Platform SONCOS, recommend timely initiation of palliative care as a standard component of comprehensive cancer care [26,27]. Studies in glioblastoma demonstrate that early ACP is associated with less aggressive end-of-life care [28] and offers important benefits for caregivers, including reduced anxiety and increased preparedness for the dying process [29]. Yet, observational studies report low rates of documented advance directives, delayed palliative care referrals, and fragmented communication between healthcare providers across specialties [28,30,31,32]. In the Netherlands, only 46% of patients had documented advance directives [33], which is concerning given the early cognitive decline affecting decision-making capacity in many patients with glioblastoma [34]. Our integrated care pathway addresses these gaps by recommending ACP initiation within six weeks of diagnosis and establishing structured coordination across settings to align care with the goals, values, and preferences of patients and their caregivers. Therefore, ACP should be offered soon after diagnosis while allowing patients and caregivers to decide when and what to discuss based on their readiness.

In integrated palliative care, the two-track approach (combining antitumor treatment and palliative care simultaneously) requires close, ongoing collaboration between professionals across care settings and disciplines. An important goal of the integrated care pathway is that the role for each clinician is clear, for example: who is the lead clinician and when, when are patients referred to specialist palliative care, and who is responsible for care coordination across sectors. Significant improvements can be made by strengthening collaboration across sectors and disciplines, ensuring that decisions are guided by what matters most to patients and their caregivers.

### 4.2. The Role of Payment Reform in Integrated Palliative Care

This study is part of a larger project that aims to develop an integrated care pathway and alternative payment model (APM) for early palliative care in patients with high-grade glioma. Currently, the pre-dominant payment model in the Netherlands, fee-for-service, rewards the volume of services delivered rather than the value or quality of care [35]. APMs offer a promising alternative by aligning reimbursement with high-quality, coordinated care [36]. APMs can link payment to specific care outcomes that are relevant for integrated palliative care, such as timely initiation of ACP, integrated care coordination, and patient- or caregiver-reported satisfaction with care.

To inform the development of the APM, we identified several perceived financial barriers to the early integration of palliative care. Some of these barriers may not reflect an actual lack of reimbursement but rather difficulties in registering and invoicing specific care activities. Notably, the barrier concerning insufficient reimbursement for ACP has been addressed by changes in the Dutch hospital payment structure. Since 2025, hospitals can claim reimbursement for ACP discussions between patient and clinician once every 365 days, provided the patient is in the palliative phase and the care plan is shared with the patient’s GP. This policy change will be incorporated in the next phase of our work, in which we will design the payment model and explore the potential for value-based payment.

### 4.3. Monitoring and Evaluation of the Integrated Care Pathway

A critical component of developing an integrated care pathway is the establishment of a robust framework for monitoring and evaluation. Successful care models require ongoing assessment of both clinical processes and patient-reported outcomes (PROs) [37]. However, selecting appropriate outcomes that reflect the value of palliative care to patients and relatives remains a challenge. Traditional indicators such as survival or healthcare utilization may not capture the benefits of early palliative care, especially in patients with progressive disease trajectories such as glioblastoma. Therefore, the use of patient-reported outcome measures (PROMs) and patient- and caregiver-reported experience measures (PREMs) is essential to track both patient’s functioning and well-being, and satisfaction with care and alignment of care with the patient’s values. Stakeholders in this study unanimously agreed that the experiences of patients and their relatives should be central for evaluating the performance of the care pathway, as well as to the evaluation of any future APM. To ensure the feasibility of monitoring, we selected nine indicators across four domains that combine PROs with routinely collected clinical data, recognizing that the latter is the most feasible and implementable in clinical practice. A comprehensive evaluation plan, including the systemic collection of indicators, should be embedded from the outset of pathway development to support continuous learning and improvement.

### 4.4. Strengths and Limitations

A major strength of this study is the involvement of healthcare providers across sectors, including hospital-based carers, general practitioners, and district nurses, enabling the development of a pathway that supports coordinated care and facilitates smoother transitions between hospital and home settings. In addition, we involved patients, caregivers, and patient advocates throughout the design process. However, our study also has limitations. First, the care pathway was developed within the Netherlands and may not be directly transferable to other countries with different healthcare systems. Nevertheless, the integrated care pathway aligns with international guidelines, such as ASCO [37], and other care pathways for glioblastoma, such as from the Cancer Council Victoria in Australia [38], particularly in its emphasis on early integration of palliative care and ACP. Second, although many barriers to early palliative care integration were addressed, some challenges remain unresolved. Notably, the fragmentation of electronic health records and digital infrastructure poses a significant barrier to effective advance care planning and palliative care coordination. Although no widespread digital infrastructure currently exists in the Netherlands for sharing care plans between primary and secondary care, such systems are under development. In the meantime, stakeholders will use a standardized form developed by the regional network for palliative care. Finally, while the pathway has been designed through an iterative process, the implementation and evaluation remain future steps. As the final step of design thinking involves real-world testing, our future work will focus on piloting the pathway and evaluating patient, caregiver and system-level outcomes of early integrated palliative care.

### 4.5. Implications for Practice

This study provides a blueprint for the systematic integration of early palliative care into routine glioblastoma management. It offers practical guidance on key components, such as the timing of ACP, interprofessional collaboration and coordination, referral criteria and triggers for specialist palliative care, and structured bereavement support. The resulting care pathway may serve as a transferable model for other institutions and healthcare systems seeking to improve palliative care delivery for patients with advanced cancer. The care pathway will be piloted and subsequently implemented in full starting in September 2026. Anticipated barriers include clinician time pressure and stigma surrounding ACP. These barriers will be addressed by providing patients and caregivers with tailored information on ACP, alongside practical support for clinicians in scheduling ACP appointments.

Several important insights emerged from the co-design process. First, the use of a design thinking approach facilitated the active engagement of a broad range of relevant stakeholders, enhancing the feasibility of implementation. Second, gaining a comprehensive understanding of barriers and unmet needs, especially from the patient and caregiver perspective, was critical to ensure that the pathway aligns with patients’ values and preferences. Third, the early involvement of healthcare administrators and health insurers proved essential for embedding the care pathway within the existing healthcare system and exploring supporting payment models for integrated care.

## 5. Conclusions

The co-designed integrated care pathway offers a feasible and scalable framework for systematically integrating early palliative care for patients with glioblastoma. The care pathway addresses clinical, organizational, and financial barriers to the early integration of palliative care and promotes coordinated, patient-centered care. Future steps include implementation; concretization in practice of the recently introduced new payment model for proactive care planning; and a robust evaluation of patient, caregiver, and system-level outcomes.

## Figures and Tables

**Figure 1 healthcare-13-02378-f001:**
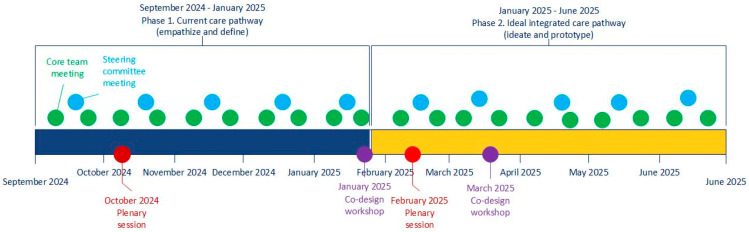
Schematic timeline of the design process with meetings and the design thinking steps.

**Figure 2 healthcare-13-02378-f002:**
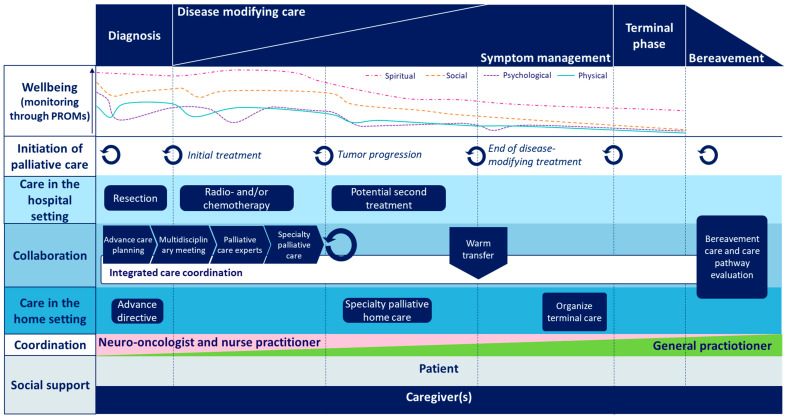
Integrated care pathway.

**Table 1 healthcare-13-02378-t001:** Overview of involved stakeholders.

Type of Stakeholders Involved	Number of Stakeholders, n	Years of Experience
*Core team*	4	
Neuro-oncologist	1	***
Neuro-oncology case manager	1	*****
Palliative care specialist and radiotherapist	1	*****
Health economist	1	*
*Steering committee*	10	
Neuro-oncologist	1	**
Specialist palliative care nurse practitioner	1	****
General practitioner and palliative care specialist	1	****
District nursing regional manager	1	*****
Patient advocate	1	**
Health economist	1	**
Regional care network coordinator	2	***
Healthcare purchaser	1	***
Healthcare administrator	1	***
Policy advisor	1	***
*Project group*	4	
Policy advisor	1	**
Healthcare purchaser	1	**
Healthcare administrator	1	***
Patient advocate	1	***
*Additional participants in informal interviews*	9	
Patient	2	
Caregiver	2	
Medical oncologist	1	
District nurse	1	
Hospice manager	1	
Nursing home physician	1	
General practitioner	1	

Years of experience: * 1–5 years, ** 5–10 years, *** 10–15 years, **** 15–20 years, ***** 20+ years.

**Table 2 healthcare-13-02378-t002:** Overview of perceived barriers to the early integration of palliative care in routine glioblastoma care.

Core Problem Statements	Perceived Barriers		
	*Clinical Practice*	*Digital Infrastructure*	*Financing*
1. ACP is not initiated or initiated late into the disease trajectory	Patients’ preferences for the timing of ACP varies; care providers may hesitate to initiate ACP; unclear referral triggers for specialist palliative care.	No digital infrastructure for sharing electronic patient files and care plans.	Insufficient reimbursement for ACP.
2. Lack of coordination between providers	Patients experience a sharp transition between hospital- and home-based care; providers may not always be familiar with glioblastoma and/or how to handle worsening symptoms.		Insufficient reimbursement for multidisciplinary meetings between primary and secondary care
3. Inappropriate care in the final phase of life	Primary care providers are not always familiar with glioma and how to handle worsening symptoms.		Fee-for-service payment model incentives volume instead of value.
4. Bereavement care is not sufficiently provided	-	-	Insufficient reimbursement for bereavement care; providing bereavement care may be discouraged by hospital management.

Abbreviations: ACP, advance care planning.

**Table 3 healthcare-13-02378-t003:** Core indicator set for future monitoring and evaluation of the integrated care pathway.

Care Pathway Goals	Indicator
**Advance care planning**	Percentage of patients and relatives who have conversations with their clinician about the final phase of life
	Place of death
	Satisfaction with care as reported by relatives (e.g., alignment between preferred and actual place of death)
**Collaboration across care sectors**	Percentage of patients discussed in multidisciplinary meetings
	Percentage of patients with a care coordinator
	Satisfaction with care coordination as reported by relatives
**Prevention of inappropriate care**	Total palliative care utilization (from diagnosis, and at 3 months and 30 days before death)
	Potentially inappropriate care in the final phase of life (3 months and 30 days before death) ≥1 emergency department visit(s) Hospital admission(s) Admission(s) longer than 14 days Chemotherapy ICU admission(s) Diagnostics, such as MRI
**Bereavement care**	Satisfaction with bereavement care as reported by relatives

## Data Availability

The data presented in this study are available on request from the corresponding author.

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
