# Peer review of "Early Palliative Care in Patients with Glioblastoma: Co-Design of an Integrated Care Pathway"

_healthcare, 2025, doi:10.3390/healthcare13182378_

Round 1
Reviewer 1 Report
Comments and Suggestions for Authors
The manuscript decribes a very interesting issue, that is the role of early palliative care in patients with glioblastoma. The study is part of a broader initiative exploring payment reform of palliative care in the Netherlands and appears to be commissioned by the Dutch Health Care Authority.
The methodology section is clear and well described. Authors choose for a design thinking approach.
Authors may consider adding a “graphical” resume (table/figure) of sections 2.3.1, 2.3.2, 2.4 to improve the quality of the manuscript.
The paper addresses an important issue, namely quality of life in diseases with a poor prognosis and patient autonomy in decision-making. Perhaps something could be added about ACP. Is it a law? A practice? A clearer explanation would help international readers.
The writing is clear, and all of the references are relevant. Nonetheless, adding the following citation would complete the reported bibliography:
Crooms, R.C.; Goldstein, N.E.; Diamond, E.L.; Vickrey, B.G. Palliative Care in High-Grade Glioma: A Review. Brain Sci. 2020, 10, 723. https://doi.org/10.3390/brainsci10100723
The topic addressed has significant implications for health policy and end-of-life matter. Therefore, I would recommend the publication of this paper after the minor revisions suggested have been made.
Author Response
Dear Reviewer 1,
Comment 1: Authors may consider adding a “graphical” resume (table/figure) of sections 2.3.1, 2.3.2, 2.4 to improve the quality of the manuscript.
Response 1: Figure 1 presents a graphical resume of sections 2.3.1, 2.3.2, and 2.4, specifically the two design thinking phases and the meetings part of the project. For clarification we now referenced Figure 1 at the start of sections 2.3.1, 2.3.2, and 2,4 [lines 132-133].
Comment 2: The paper addresses an important issue, namely quality of life in diseases with a poor prognosis and patient autonomy in decision-making. Perhaps something could be added about ACP. Is it a law? A practice? A clearer explanation would help international readers.
Response 2: Thank you for pointing this out. We have included the full definition of advance care planning as developed by Sudore, et al. in the introduction of our manuscript [lines 57-60].
Sudore RL, Lum HD, You JJ, Hanson LC, Meier DE, Pantilat SZ, et al. Defining Advance Care Planning for Adults: A Consensus Definition From a Multidisciplinary Delphi Panel. J Pain Symptom Manage. 2017;53(5):821-32 e1
Comment 3: The writing is clear, and all of the references are relevant. Nonetheless, adding the following citation would complete the reported bibliography:
Crooms, R.C.; Goldstein, N.E.; Diamond, E.L.; Vickrey, B.G. Palliative Care in High-Grade Glioma: A Review. Brain Sci. 2020, 10, 723. https://doi.org/10.3390/brainsci10100723
Response 3: We have included this reference in the manuscript [line 54].
Reviewer 2 Report
Comments and Suggestions for Authors
This is a timely and well-structured manuscript that addresses the critical need for early integration of palliative care in glioblastoma management. The use of a design thinking and co-creation approach with diverse stakeholders is innovative and highly relevant. The paper is clear, comprehensive, and contributes valuable insights to both clinical practice and health policy.
Major Points
- Generalizability: The pathway is developed within the Dutch healthcare system; more discussion on applicability to other contexts would be useful.
- Implementation: Strategies for piloting and real-world testing of the pathway should be further elaborated, including expected barriers and evaluation metrics.
- Digital infrastructure: The lack of integrated electronic records is identified as a barrier; possible interim solutions or mitigations should be discussed.
- Patient and caregiver involvement: Their input is valuable but appears limited; a clearer account of their influence on the final design would strengthen the paper.
- Indicators: The proposed monitoring framework may be complex; prioritizing key, feasible indicators would improve usability in clinical practice.
Minor Points
- Some sections of the results are quite detailed; a more concise summary of the main findings could improve readability.
- Figures and tables are informative, but Figure 2 could benefit from a clearer visual separation of hospital-based and home-based care elements.
- The discussion could be enriched by briefly comparing the proposed pathway with existing international models of early palliative care integration in oncology.
- Stylistic/formatting: a few references (e.g., "[ref]") appear as placeholders and should be completed before publication.
- Consider clarifying terminology: terms such as "transmural care" may not be widely familiar to an international readership; a brief definition would be helpful.
Recommendation: Overall, this manuscript presents a significant and well-executed contribution. I recommend “acceptance after minor revisions”, to refine implementation details and strengthen the discussion of applicability beyond the Dutch context.
Author Response
Dear Reviewer 2,
Major Points
Comment 1: Generalizability: The pathway is developed within the Dutch healthcare system; more discussion on applicability to other contexts would be useful.
Response 1: Thank you for this helpful suggestions. We have revised the manuscript accordingly and included a discussion on the adaptability in the discussion [lines 396-399].
Comment 2: Implementation: Strategies for piloting and real-world testing of the pathway should be further elaborated, including expected barriers and evaluation metrics.
Response 2: We have included details on the implementation, potential barriers and how we plan to overcome these in the discussion [lines 418-421].
Comment 3: Digital infrastructure: The lack of integrated electronic records is identified as a barrier; possible interim solutions or mitigations should be discussed.
Response 3: We have added interim solutions to the lack of a central digital infrastructure to the discussion [lines 403-406].
Comment 4: Patient and caregiver involvement: Their input is valuable but appears limited; a clearer account of their influence on the final design would strengthen the paper.
Response 4: We have added an additional paragraph to the methods section to describe how we contacted patients and caregivers and how they were involved [lines 144-160]. In addition, we clarified in the methods section that at least one patient advocate was present for all meetings [lines 110-112].
Comment 5: Indicators: The proposed monitoring framework may be complex; prioritizing key, feasible indicators would improve usability in clinical practice.
Response 5: To ensure feasibility, we opted to select only nine indicators in four domains for monitoring the integrated care pathway. We do recognize that this still may be complex and, therefore, added a further prioritization to the discussion [lines 383-385].
Minor Points
Comment 6: Some sections of the results are quite detailed; a more concise summary of the main findings could improve readability.
Response 6: We now provide a summary of main findings in the first paragraph of the discussion section [line 311-326].
Comment 7: Figures and tables are informative, but Figure 2 could benefit from a clearer visual separation of hospital-based and home-based care elements.
Response 7: The palliative care elements can take place in both a hospital-based and home-based setting depending on the disease trajectory of the patient, and are, therefore, presented separately. To clarify that this is the case, we have added an explanation to the results section discussing these palliative care elements [lines 282-285].
Comment 8: The discussion could be enriched by briefly comparing the proposed pathway with existing international models of early palliative care integration in oncology.
Response 8: We have revised the manuscript accordingly and included a comparison to other oncological care models in the discussion [lines 396-399].
Comment 9: Stylistic/formatting: a few references (e.g., "[ref]") appear as placeholders and should be completed before publication.
Response 9: we have replaced the placeholders with the appropriate references.
Comment 10: Consider clarifying terminology: terms such as "transmural care" may not be widely familiar to an international readership; a brief definition would be helpful.
Response 10: Thank you for this suggestion. ‘Transmural care’ is indeed a Dutch term meaning simply integrated care. To avoid confusion, we have removed ‘transmural care’ from the manuscript and use ‘integrated care’ or ‘collaboration across sectors’ instead.
Reviewer 3 Report
Comments and Suggestions for Authors
This study focuses on the systematic integration of early palliative care in glioblastoma management and addresses a significant gap. The design-focused thinking approach is innovative and valuable. The article presents a well-structured, methodologically transparent, and practically applicable model.
1) The selection of patients and caregivers, how they were invited, and their representativeness should be more clearly defined.
2) Participants' professional experience/years of practice should be listed in Table 1. This is important for the representativeness of the group.
3) The analysis of data obtained from interviews, including interview notes and field notes, should be explained more systematically. Direct quotes from patients and caregivers or short case examples are recommended. Was software used for analysis?
4) The "testing" phase, one of the five stages of design-focused thinking, was not included in this study. This was left for future work; the design was not tested in a pilot study.
Comments regarding the author's self-citation references:
The sentence derived from sources 7 and 8 is as follows:
"Given the progressive and incurable nature of glioblastoma, the integration of early palliative care into routine care is essential to support treatment decision-making and optimal symptom management (7, 8)"
The authors in the article cited previously published articles. This leads to an unwarranted citation enrichment and reinforces the idea of ghostwriting. I have already suggested a major revision to this article.
In addition to the suggestion, I suggest that the authors "remove the citations of names mentioned in the article."
Author Response
Dear Reviewer 3,
Comment 1: The selection of patients and caregivers, how they were invited, and their representativeness should be more clearly defined.
Response 1: We have added an additional paragraph to the methods section to describe how we contacted patients and caregivers and how they were involved [lines 144-160]. In addition, we clarified in the methods section that at least one patient advocate was present for all meetings [lines 110-112].
Comment 2: Participants' professional experience/years of practice should be listed in Table 1. This is important for the representativeness of the group.
Response 2: We have revised the manuscript accordingly and included the years of experience/practice in table 1 in five categories: 1-5 years, 5-10 years. 10-15 years, 15-20 years and 20+ years [lines 115-116].
Comment 3: The analysis of data obtained from interviews, including interview notes and field notes, should be explained more systematically. Direct quotes from patients and caregivers or short case examples are recommended. Was software used for analysis?
Response 3: We have added a short paragraph to the methods section to detail qualitative data collection and analysis [lines 161-165].
Comment 4: The "testing" phase, one of the five stages of design-focused thinking, was not included in this study. This was left for future work; the design was not tested in a pilot study.
Response 4: The testing phase indeed falls out of the scope of this manuscript. We have added details on the pilot and implementation phase to the discussion [lines 418-421].
Comment 5: Comments regarding the author's self-citation references:
The sentence derived from sources 7 and 8 is as follows:
"Given the progressive and incurable nature of glioblastoma, the integration of early palliative care into routine care is essential to support treatment decision-making and optimal symptom management (7, 8)"
The authors in the article cited previously published articles. This leads to an unwarranted citation enrichment and reinforces the idea of ghostwriting. I have already suggested a major revision to this article.
In addition to the suggestion, I suggest that the authors "remove the citations of names mentioned in the article."
Response 5: We thank the reviewer for pointing this out. We have thoroughly reviewed the list of references and excluded all unnecessary self-citations. Citations of previous work now only include references essential to the development of palliative care and advance care planning in neuro-oncology, or references to questionnaires (e.g. EORTC QLQ-BN20) which are used for symptom monitoring within the care pathway.
Round 2
Reviewer 3 Report
Comments and Suggestions for Authors
Corrections made by the author are acceptable.